# Sinkhorn Distances:
# Lightspeed Computation of Optimal Transport

**Marco Cuturi**
Graduate School of Informatics, Kyoto University
`mcuturi@i.kyoto-u.ac.jp`

## Abstract

Optimal transport distances are a fundamental family of distances for probability measures and histograms of features. Despite their appealing theoretical properties, excellent performance in retrieval tasks and intuitive formulation, their computation involves the resolution of a linear program whose cost can quickly become prohibitive whenever the size of the support of these measures or the histograms' dimension exceeds a few hundred. We propose in this work a new family of optimal transport distances that look at transport problems from a maximum-entropy perspective. We smooth the classic optimal transport problem with an entropic regularization term, and show that the resulting optimum is also a distance which can be computed through Sinkhorn's matrix scaling algorithm at a speed that is several orders of magnitude faster than that of transport solvers. We also show that this regularized distance improves upon classic optimal transport distances on the MNIST classification problem.

## 1 Introduction

Choosing a suitable distance to compare probabilities is a key problem in statistical machine learning. When little is known on the probability space on which these probabilities are supported, various information divergences with minimalistic assumptions have been proposed to play that part, among which the Hellinger, $\chi_2$, total variation or Kullback-Leibler divergences. When the probability space is a metric space, optimal transport distances (Villani, 2009, §6), *a.k.a.* earth mover's (EMD) in computer vision (Rubner et al., 1997), define a more powerful geometry to compare probabilities.

This power comes, however, with a heavy computational price tag. No matter what the algorithm employed – network simplex or interior point methods – the cost of computing optimal transport distances scales at least in $O(d^3 log(d))$ when comparing two histograms of dimension $d$ or two point clouds each of size $d$ in a general metric space (Pele and Werman, 2009, §2.1).

In the particular case that the metric probability space of interest can be embedded in $\mathbb{R}^n$ and $n$ is small, computing or approximating optimal transport distances can become reasonably cheap. Indeed, when $n = 1$, their computation only requires $O(d \log d)$ operations. When $n \geq 2$, embeddings of measures can be used to approximate them in linear time (Indyk and Thaper, 2003; Grauman and Darrell, 2004; Shirdhonkar and Jacobs, 2008) and network simplex solvers can be modified to run in quadratic time (Gudmundsson et al., 2007; Ling and Okada, 2007). However, the distortions of such embeddings (Naor and Schechtman, 2007) as well as the exponential increase of costs incurred by such modifications as $n$ grows make these approaches inapplicable when $n$ exceeds 4. Outside of the perimeter of these cases, computing a single distance between a pair of measures supported by a few hundred points/bins in an arbitrary metric space can take more than a few seconds on a single CPU. This issue severely hinders the applicability of optimal transport distances in large-scale data analysis and goes as far as putting into question their relevance within the field of machine learning, where high-dimensional histograms and measures in high-dimensional spaces are now prevalent.

We show in this paper that another strategy can be employed to speed-up optimal transport, and even potentially define a better distance in inference tasks. Our strategy is valid regardless of the metric characteristics of the original probability space. Rather than exploit properties of the metric probability space of interest (such as embeddability in a low-dimensional Euclidean space) we choose to focus directly on the original transport problem, and regularize it with an entropic term. We argue that this regularization is intuitive given the geometry of the optimal transport problem and has, in fact, been long known and favored in transport theory to predict traffic patterns (Wilson, 1969). From an optimization point of view, this regularization has multiple virtues, among which that of turning the transport problem into a strictly convex problem that can be solved with matrix scaling algorithms. Such algorithms include Sinkhorn's celebrated fixed point iteration (1967), which is known to have a linear convergence (Franklin and Lorenz, 1989; Knight, 2008). Unlike other iterative simplex-like methods that need to cycle through complex conditional statements, the execution of Sinkhorn's algorithm only relies on matrix-vector products. We propose a novel implementation of this algorithm that can compute *simultaneously* the distance of a single point to a family of points using matrix-matrix products, and which can therefore be implemented on GPGPU architectures. We show that, on the benchmark task of classifying MNIST digits, regularized distances perform better than standard optimal transport distances, and can be computed several orders of magnitude faster.

This paper is organized as follows: we provide reminders on optimal transport theory in Section 2, introduce Sinkhorn distances in Section 3 and provide algorithmic details in Section 4. We follow with an empirical study in Section 5 before concluding.

## 2 Reminders on Optimal Transport

**Transport Polytope and Interpretation as a Set of Joint Probabilities**    In what follows, $\langle \cdot, \cdot \rangle$ stands for the Frobenius dot-product. For two probability vectors $r$ and $c$ in the simplex $\Sigma_d := \{x \in \mathbb{R}_+^d : x^T \mathbf{1}_d = 1\}$, where $\mathbf{1}_d$ is the $d$ dimensional vector of ones, we write $U(r,c)$ for the transport polytope of $r$ and $c$, namely the polyhedral set of $d \times d$ matrices,

$$U(r,c) := \{P \in \mathbb{R}_+^{d \times d} \mid P\mathbf{1}_d = r, P^T\mathbf{1}_d = c\}.$$

$U(r,c)$ contains all nonnegative $d \times d$ matrices with row and column sums $r$ and $c$ respectively. $U(r,c)$ has a probabilistic interpretation: for $X$ and $Y$ two multinomial random variables taking values in $\{1, \cdots, d\}$, each with distribution $r$ and $c$ respectively, the set $U(r,c)$ contains all possible *joint probabilities* of $(X,Y)$. Indeed, any matrix $P \in U(r,c)$ can be identified with a joint probability for $(X,Y)$ such that $p(X = i, Y = j) = p_{ij}$. We define the entropy $h$ and the Kullback-Leibler divergences of $P, Q \in U(r,c)$ and a marginals $r \in \Sigma_d$ as

$$h(r) = -\sum_{i=1}^{d} r_i \log r_i, \quad h(P) = -\sum_{i,j=1}^{d} p_{ij} \log p_{ij}, \quad \mathbf{KL}(P\|Q) = \sum_{ij} p_{ij} \log \frac{p_{ij}}{q_{ij}}.$$

**Optimal Transport Distance Between $r$ and $c$**    Given a $d \times d$ cost matrix $M$, the cost of mapping $r$ to $c$ using a transport matrix (or joint probability) $P$ can be quantified as $\langle P, M \rangle$. The problem defined in Equation (1)

$$d_M(r,c) := \min_{P \in U(r,c)} \langle P, M \rangle. \tag{1}$$

is called an *optimal transport (OT)* problem between $r$ and $c$ given cost $M$. An optimal table $P^\star$ for this problem can be obtained, among other approaches, with the network simplex (Ahuja et al., 1993, §9). The optimum of this problem, $d_M(r,c)$, is a distance between $r$ and $c$ (Villani, 2009, §6.1) whenever the matrix $M$ is itself a metric matrix, namely whenever $M$ belongs to the cone of distance matrices (Avis, 1980; Brickell et al., 2008):

$$\mathcal{M} = \{M \in \mathbb{R}_+^{d \times d} : \forall i, j \leq d, m_{ij} = 0 \Leftrightarrow i = j, \ \forall i, j, k \leq d, m_{ij} \leq m_{ik} + m_{kj}\}.$$

For a general matrix $M$, the worst case complexity of computing that optimum scales in $O(d^3 \log d)$ for the best algorithms currently proposed, and turns out to be super-cubic in practice as well (Pele and Werman, 2009, §2.1).

## 3   Sinkhorn Distances: Optimal Transport with Entropic Constraints

**Entropic Constraints on Joint Probabilities**   The following information theoretic inequality (Cover and Thomas, 1991, §2) for joint probabilities

$$\forall r, c \in \Sigma_d, \forall P \in U(r,c), h(P) \leq h(r) + h(c),$$

is tight, since the *independence table* $rc^T$ (Good, 1963) has entropy $h(rc^T) = h(r) + h(c)$. By the concavity of entropy, we can introduce the convex set

$$U_\alpha(r,c) := \{P \in U(r,c) \,|\, \mathbf{KL}(P \| rc^T) \leq \alpha\} = \{P \in U(r,c) \,|\, h(P) \geq h(r) + h(c) - \alpha\} \subset U(r,c).$$

These two definitions are indeed equivalent, since one can easily check that $\mathbf{KL}(P\|rc^T) = h(r) + h(c) - h(P)$, a quantity which is also the mutual information $I(X\|Y)$ of two random variables $(X, Y)$ should they follow the joint probability $P$ (Cover and Thomas, 1991, §2). Hence, the set of tables $P$ whose Kullback-Leibler divergence to $rc^T$ is constrained to lie below a certain threshold can be interpreted as the set of joint probabilities $P$ in $U(r,c)$ which have *sufficient entropy* with respect to $h(r)$ and $h(c)$, or small enough *mutual information*. For reasons that will become clear in Section 4, we call the quantity below the Sinkhorn distance of $r$ and $c$:

**Definition 1** (Sinkhorn Distance). $d_{M,\alpha}(r,c) := \min\limits_{P \in U_\alpha(r,c)} \langle P, M \rangle$

Why consider an entropic constraint in optimal transport? The first reason is computational, and is detailed in Section 4. The second reason is built upon the following intuition. As a classic result of linear optimization, the OT problem is always solved on a vertex of $U(r,c)$. Such a vertex is a sparse $d \times d$ matrix with only up to $2d - 1$ non-zero elements (Brualdi, 2006, §8.1.3). From a probabilistic perspective, such vertices are quasi-deterministic joint probabilities, since if $p_{ij} > 0$, then very few probabilities $p_{ij'}$ for $j \neq j'$ will be non-zero in general. Rather than considering such outliers of $U(r,c)$ as the basis of OT distances, we propose to restrict the search for low cost joint probabilities to tables with sufficient smoothness. Note that this is equivalent to considering the maximum-entropy principle (Jaynes, 1957; Darroch and Ratcliff, 1972) if we were to maximize entropy while keeping the transportation cost constrained.

Before proceeding to the description of the properties of Sinkhorn distances, we note that Ferradans et al. (2013) have recently explored similar ideas. They relax and penalize (through graph-based norms) the original transport problem to avoid undesirable properties exhibited by the original optima in the problem of color matching. Combined, their idea and ours suggest that many more smooth regularizers will be worth investigating to solve the the OT problem, driven by either or both computational and modeling motivations.

**Metric Properties of Sinkhorn Distances**   When $\alpha$ is large enough, the Sinkhorn distance coincides with the classic OT distance. When $\alpha = 0$, the Sinkhorn distance has a closed form and becomes a negative definite kernel if one assumes that $M$ is itself a negative definite distance, or equivalently a Euclidean distance matrix[1].

**Property 1.** *For $\alpha$ large enough, the Sinkhorn distance $d_{M,\alpha}$ is the transport distance $d_M$.*

*Proof.* Since for any $P \in U(r,c), h(P)$ is lower bounded by $\frac{1}{2}(h(r) + h(c))$, we have that for $\alpha$ large enough $U_\alpha(r,c) = U(r,c)$ and thus both quantities coincide.■

**Property 2** (Independence Kernel). $d_{M,0} = r^T M c$. *If $M$ is a Euclidean distance matrix, $d_{M,0}$ is a negative definite kernel and $e^{-t d_{M,0}}$, the independence kernel, is positive definite for all $t > 0$.*

The proof is provided in the appendix. Beyond these two extreme cases, the main theorem of this section states that Sinkhorn distances are symmetric and satisfy triangle inequalities for all possible values of $\alpha$. Since for $\alpha$ small enough $d_{M,\alpha}(r,r) > 0$ for any $r$ such that $h(r) > 0$, Sinkhorn distances cannot satisfy the *coincidence axiom* ($d(x,y) = 0 \Leftrightarrow x = y$ holds for all $x, y$). However, multiplying $d_{M,\alpha}$ by $\mathbf{1}_{r \neq c}$ suffices to recover the coincidence property if needed.

**Theorem 1.** *For all $\alpha \geq 0$ and $M \in \mathcal{M}$, $d_{M,\alpha}$ is symmetric and satisfies all triangle inequalities. The function $(r,c) \mapsto \mathbf{1}_{r \neq c} d_{M,\alpha}(r,c)$ satisfies all three distance axioms.*

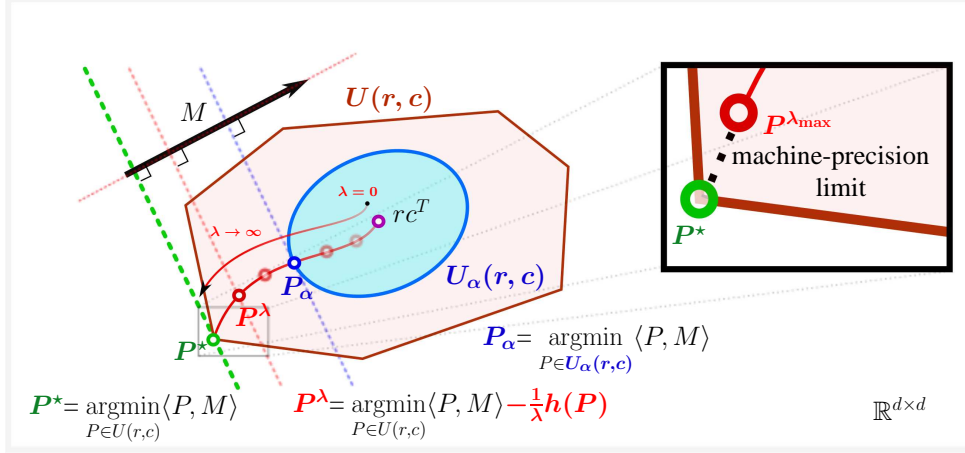

Figure 1: Transport polytope $U(r, c)$ and Kullback-Leibler ball $U_\alpha(r, c)$ of level $\alpha$ centered around $rc^T$. This drawing implicitly assumes that the optimal transport $P^\star$ is unique. The Sinkhorn distance $d_{M,\alpha}(r, c)$ is equal to $\langle P_\alpha, M \rangle$, the minimum of the dot product with $M$ on that ball. For $\alpha$ large enough, both objectives coincide, as $U_\alpha(r, c)$ gradually overlaps with $U(r, c)$ in the vicinity of $P^\star$. The dual-sinkhorn distance $d_M^\lambda(r, c)$, the minimum of the transport problem regularized by minus the entropy divided by $\lambda$, reaches its minimum at a unique solution $P^\lambda$, forming a regularization path for varying $\lambda$ from $rc^T$ to $P^\star$. For a given value of $\alpha$, and a pair $(r, c)$ there exists $\lambda \in [0, \infty]$ such that both $d_M^\lambda(r, c)$ and $d_{M,\alpha}(r, c)$ coincide. $d_M^\lambda$ can be efficiently computed using Sinkhorn's fixed point iteration (1967). Although the convergence to $P^\star$ of this fixed point iteration is theoretically guaranteed as $\lambda \to \infty$, the procedure cannot work beyond a problem-dependent value $\lambda_{\max}$ beyond which some entries of $e^{-\lambda M}$ are represented as zeroes in memory.

The gluing lemma (Villani, 2009, p.19) is key to proving that OT distances are indeed distances. We propose a variation of this lemma to prove our result:

**Lemma 1** (Gluing Lemma With Entropic Constraint)**.** *Let $\alpha \geq 0$ and $x, y, z \in \Sigma_d$. Let $P \in U_\alpha(x, y)$ and $Q \in U_\alpha(y, z)$. Let $S$ be the $d \times d$ defined as $s_{ik} := \sum_j \frac{p_{ij} q_{jk}}{y_j}$. Then $S \in U_\alpha(x, z)$.*

The proof is provided in the appendix. We can prove the triangle inequality for $d_{M,\alpha}$ by using the same proof strategy than that used for classic transport distances:

*Proof of Theorem 1.* The symmetry of $d_{M,\alpha}$ is a direct result of $M$'s symmetry. Let $x, y, z$ be three elements in $\Sigma_d$. Let $P \in U_\alpha(x, y)$ and $Q \in U_\alpha(y, z)$ be two optimal solutions for $d_{M,\alpha}(x, y)$ and $d_{M,\alpha}(y, z)$ respectively. Using the matrix $S$ of $U_\alpha(x, z)$ provided in Lemma 1, we proceed with the following chain of inequalities:

$$d_{M,\alpha}(x, z) = \min_{P \in U_\alpha(x,z)} \langle P, M \rangle \leq \langle S, M \rangle = \sum_{ik} m_{ik} \sum_j \frac{p_{ij} q_{jk}}{y_j} \leq \sum_{ijk} (m_{ij} + m_{jk}) \frac{p_{ij} q_{jk}}{y_j}$$

$$= \sum_{ijk} m_{ij} \frac{p_{ij} q_{jk}}{y_j} + m_{jk} \frac{p_{ij} q_{jk}}{y_j} = \sum_{ij} m_{ij} p_{ij} \sum_k \frac{q_{jk}}{y_j} + \sum_{jk} m_{jk} q_{jk} \sum_i \frac{p_{ij}}{y_j}$$

$$= \sum_{ij} m_{ij} p_{ij} + \sum_{jk} m_{jk} q_{jk} = d_{M,\alpha}(x, y) + d_{M,\alpha}(y, z). \blacksquare$$

# 4 Computing Regularized Transport with Sinkhorn's Algorithm

We consider in this section a Lagrange multiplier for the entropy constraint of Sinkhorn distances:

$$\text{For } \lambda > 0, \; d_M^\lambda(r, c) := \langle P^\lambda, M \rangle, \; \text{where } P^\lambda = \operatorname*{argmin}_{P \in U(r,c)} \langle P, M \rangle - \frac{1}{\lambda} h(P). \tag{2}$$

By duality theory we have that to each $\alpha$ corresponds a $\lambda \in [0, \infty]$ such that $d_{M,\alpha}(r, c) = d_M^\lambda(r, c)$ holds for that pair $(r, c)$. We call $d_M^\lambda$ the *dual-Sinkhorn divergence* and show that it can be computed

for a much cheaper cost than the original distance $d_M$. Figure 1 summarizes the relationships between $d_M$, $d_{M,\alpha}$ and $d_M^\lambda$. Since the entropy of $P^\lambda$ decreases monotonically with $\lambda$, computing $d_{M,\alpha}$ can be carried out by computing $d_M^\lambda$ with increasing values of $\lambda$ until $h(P^\lambda)$ reaches $h(r)+h(c)-\alpha$. We do not consider this problem here and only use the dual-Sinkhorn divergence in our experiments.

**Computing $d_M^\lambda$ with Matrix Scaling Algorithms**    Adding an entropy regularization to the optimal transport problem enforces a simple structure on the optimal regularized transport $P^\lambda$:

**Lemma 2.** *For $\lambda > 0$, the solution $P^\lambda$ is unique and has the form $P^\lambda = \mathbf{diag}(u)K\,\mathbf{diag}(v)$, where $u$ and $v$ are two non-negative vectors of $\mathbb{R}^d$ uniquely defined up to a multiplicative factor and $K := e^{-\lambda M}$ is the element-wise exponential of $-\lambda M$.*

*Proof.* The existence and unicity of $P^\lambda$ follows from the boundedness of $U(r,c)$ and the strict convexity of minus the entropy. The fact that $P^\lambda$ can be written as a rescaled version of $K$ is a well known fact in transport theory (Erlander and Stewart, 1990, §3.3): let $\mathcal{L}(P,\alpha,\beta)$ be the Lagrangian of Equation (2) with dual variables $\alpha,\beta \in \mathbb{R}^d$ for the two equality constraints in $U(r,c)$:

$$\mathcal{L}(P,\alpha,\beta) = \sum_{ij} \frac{1}{\lambda} p_{ij} \log p_{ij} + p_{ij} m_{ij} + \alpha^T(P\mathbf{1}_d - r) + \beta^T(P^T\mathbf{1}_d - c).$$

For any couple $(i,j)$, $(\partial \mathcal{L}/\partial p_{ij} = 0) \Rightarrow p_{ij} = e^{-1/2-\lambda\alpha_i} e^{-\lambda m_{ij}} e^{-1/2-\lambda\beta_j}$. Since $K$ is strictly positive, Sinkhorn's theorem (1967) states that there exists a *unique matrix* of the form $\mathbf{diag}(u)K\,\mathbf{diag}(v)$ that belongs to $U(r,c)$, where $u,v \geq \mathbf{0}_d$. $P^\lambda$ is thus necessarily that matrix, and can be computed with Sinkhorn's fixed point iteration $(u,v) \leftarrow (r./Kv, c./K'u)$. ∎

Given $K$ and marginals $r$ and $c$, one only needs to iterate Sinkhorn's update a sufficient number of times to converge to $P^\lambda$. One can show that these successive updates carry out iteratively the projection of $K$ on $U(r,c)$ in the Kullback-Leibler sense. This fixed point iteration can be written as a single update $u \leftarrow r./K(c./K'u)$. When $r > \mathbf{0}_d$, $\mathbf{diag}(1./r)K$ can be stored in a $d \times d$ matrix $\tilde{K}$ to save one Schur vector product operation with the update $u \leftarrow 1./(\tilde{K}(c./K'u))$. This can be easily ensured by selecting the positive indices of $r$, as seen in the first line of Algorithm 1.

---

**Algorithm 1** Computation of $\mathbf{d} = [d_M^\lambda(r,c_1), \cdots, d_M^\lambda(r,c_N)]$, using Matlab syntax.

**Input** $M,\lambda,r,C := [c_1, \cdots, c_N]$.
$I = (r > 0); r = r(I); M = M(I,:); K = \exp(-\lambda M)$
$u = \texttt{ones}(\texttt{length}(r),N)/\texttt{length}(r);$
$\tilde{K} = \texttt{bsxfun}(\texttt{@rdivide},K,r)$ % equivalent to $\tilde{K} = \mathbf{diag}(1./r)K$
**while** $u$ changes or any other relevant stopping criterion **do**
    $u = 1./(\tilde{K}(C./(K'u)))$
**end while**
$v = C./(K'u)$
$\mathbf{d} = \texttt{sum}(u.*((K.*M)v))$

---

**Parallelism, Convergence and Stopping Criteria**    As can be seen right above, Sinkhorn's algorithm can be vectorized and generalized to $N$ target histograms $c_1, \cdots, c_N$. When $N = 1$ and $C$ is a vector in Algorithm 1, we recover the simple iteration mentioned in the proof of Lemma 2. When $N > 1$, the computations for $N$ target histograms can be simultaneously carried out by updating a single matrix of scaling factors $u \in \mathbb{R}_+^{d \times N}$ instead of updating a scaling vector $u \in \mathbb{R}_+^d$. This important observation makes the execution of Algorithm 1 particularly suited to GPGPU platforms. Despite ongoing research in that field (Bieling et al., 2010) such speed ups have not been yet achieved on complex iterative procedures such as the network simplex. Using Hilbert's projective metric, Franklin and Lorenz (1989) prove that the convergence of the scaling factor $u$ (as well as $v$) is linear, with a rate bounded above by $\kappa(K)^2$, where

$$\kappa(K) = \frac{\sqrt{\theta(K)}-1}{\sqrt{\theta(K)}+1} < 1, \text{ and } \theta(K) = \max_{i,j,l,m} \frac{K_{il}K_{jm}}{K_{jl}K_{im}}.$$

The upper bound $\kappa(K)$ tends to 1 as $\lambda$ grows, and we do observe a slower convergence as $P^\lambda$ gets closer to the optimal vertex $P^\star$ (or the optimal facet of $U(r,c)$ if it is not unique). Different stopping criteria can be used for Algorithm 1. We consider two in this work, which we detail below.

# 5 Experimental Results

**MNIST Digits** We test the performance of dual-Sinkhorn divergences on the MNIST digits dataset. Each image is converted to a vector of intensities on the $20 \times 20$ pixel grid, which are then normalized to sum to 1. We consider a subset of $N \in \{3, 5, 12, 17, 25\} \times 10^3$ points in the dataset. For each subset, we provide mean and standard deviation of classification error using a 4 fold (3 test, 1 train) cross validation (CV) scheme repeated 6 times, resulting in 24 different experiments. Given a distance $d$, we form the kernel $e^{-d/t}$, where $t > 0$ is chosen by CV on each training fold within $\{1, q_{10}(d), q_{20}(d), q_{50}(d)\}$, where $q_s$ is the $s\%$ quantile of a subset of distances observed in that fold. We regularize non-positive definite kernel matrices resulting from this computation by adding a sufficiently large diagonal term. SVM's were run with Libsvm (one-vs-one) for multiclass classification. We select the regular-

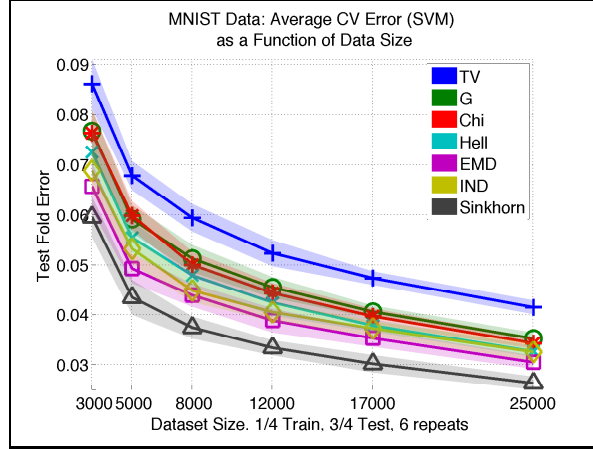

Figure 2: Average test errors with shaded confidence intervals. Errors are computed using 1/4 of the dataset for train and 3/4 for test. Errors are averaged over 4 folds × 6 repeats = 24 experiments.

ization $C$ in $10^{\{-2,0,4\}}$ using 2 folds/2 repeats CV on the training fold. We consider the Hellinger, $\chi_2$, total variation and squared Euclidean (Gaussian kernel) distances. $M$ is the $400 \times 400$ matrix of Euclidean distances between the $20 \times 20$ bins in the grid. We also tried Mahalanobis distances on this example using `exp(-tM.^2)`, `t>0` as well as its inverse, with varying values of $t$, but none of these results proved competitive. For the Independence kernel we considered $[m_{ij}^a]$ where $a \in \{0.01, 0.1, 1\}$ is chosen by CV on each training fold. We select $\lambda$ in $\{5, 7, 9, 11\} \times 1/q_{50}(M)$ where $q_{50}(M(:))$ is the median distance between pixels. We set the number of fixed-point iterations to an arbitrary number of 20 iterations. In most (though not all) folds, the value $\lambda = 9$ comes up as the best setting. The dual-Sinkhorn divergence beats by a safe margin all other distances, including the classic optimal transport distance, here labeled as EMD.

**Does the Dual-Sinkhorn Divergence Converge to the EMD?** We study the convergence of the dual-Sinkhorn divergence towards classic optimal transport distances as $\lambda$ grows. Because of the regularization in Equation (2), $d_M^\lambda(r, c)$ is necessarily larger than $d_M(r, c)$, and we expect this gap to decrease as $\lambda$ increases. Figure 3 illustrates this by plotting the boxplot of the distributions of $(d_M^\lambda(r, c) - d_M(r, c))/d_M(r, c)$ over $40^2$ pairs of images from the MNIST database. $d_M^\lambda$ typically approximates the EMD with a high accuracy when $\lambda$ exceeds 50 (median relative gap of 3.4% and 1.2% for 50 and 100 respectively). For this experiment as well as *all the other experiments below*, we compute a vector of $N$ divergences $\mathbf{d}$ at each iteration, and stop when *none* of the $N$ values of $\mathbf{d}$ varies more in absolute value than a 1/100th of a percent *i.e.* we stop when $\|\mathbf{d_t}./\mathbf{d}_{t-1} - 1\|_\infty < 1e - 4$.

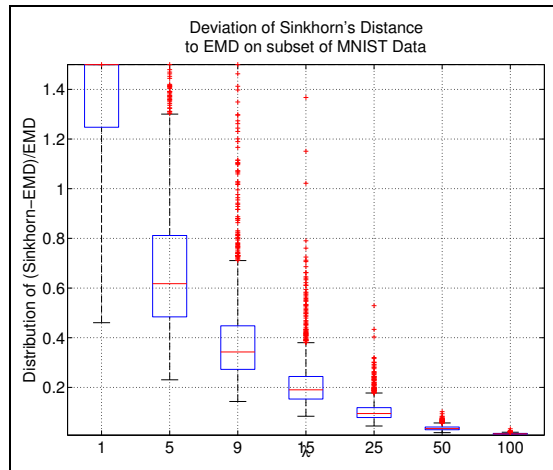

Figure 3: Decrease of the gap between the dual-Sinkhorn divergence and the EMD as a function of $\lambda$ on a subset of the MNIST dataset.

**Several Orders of Magnitude Faster**
We measure the computational speed of classic optimal transport distances vs. that of dual-Sinkhorn divergences using Rubner et al.'s (1997) and Pele and Werman's (2009) publicly available implementations. We pick a random distance matrix $M$ by generating a random graph of $d$ vertices with edge presence probability $1/2$ and edge weights uniformly distributed between 0 and 1. $M$ is the all-pairs shortest-path matrix obtained from this connectivity matrix using the Floyd-Warshall algorithm (Ahuja et al., 1993, §5.6). Using this procedure, $M$ is likely to be an extreme ray of the cone $\mathcal{M}$ (Avis, 1980, p.138). The elements of $M$ are then normalized to have unit median. We implemented Algorithm 1 in matlab, and use `emd_mex` and `emd_hat_gd_metric` mex/C files. The EMD distances and Sinkhorn CPU are run on a single core (2.66 Ghz Xeon). Sinkhorn GPU is run on a NVidia Quadro K5000 card. We consider $\lambda$ in $\{1, 10, 50\}$. $\lambda = 1$ results in a relatively dense matrix $K$, with results

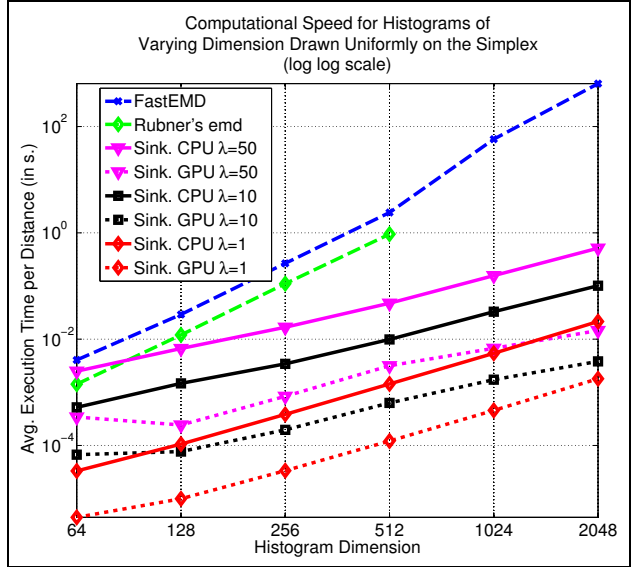

Figure 4: Average computational time required to compute a distance between two histograms sampled uniformly in the $d$ dimensional simplex for varying values of $d$. Dual-Sinkhorn divergences are run both on a single CPU and on a GPU card.

comparable to that of the Independence kernel, while for $\lambda = 10$ or $50$ $K = e^{-\lambda M}$ has very small values. Rubner et al.'s implementation cannot be run for histograms larger than $d = 512$. As can be expected, the competitive advantage of dual-Sinkhorn divergences over EMD solvers increases with the dimension. Using a GPU results in a speed-up of an additional order of magnitude.

**Empirical Complexity** To provide an accurate picture of the actual cost of the algorithm, we replicate the experiments above but focus now on the number of iterations (matrix-matrix products) typically needed to obtain the convergence of a set of $N$ divergences from a given point $r$, all uniformly sampled on the simplex. As can be seen in Figure 5, the number of iterations required for vector $\mathbf{d}$ to converge increases as $e^{-\lambda M}$ becomes diagonally dominant. However, the total number of iterations does not seem to vary with respect to the dimension. This observation can explain why we do observe a quadratic (empirical) time complexity $O(d^2)$ with respect to the dimension $d$ in Figure 4 above. These results suggest that the costly action of keeping track of the actual approximation error (computing variations in $\mathbf{d}$) is not required, and that simply predefining a fixed number of iterations can work well and yield even additional speedups.

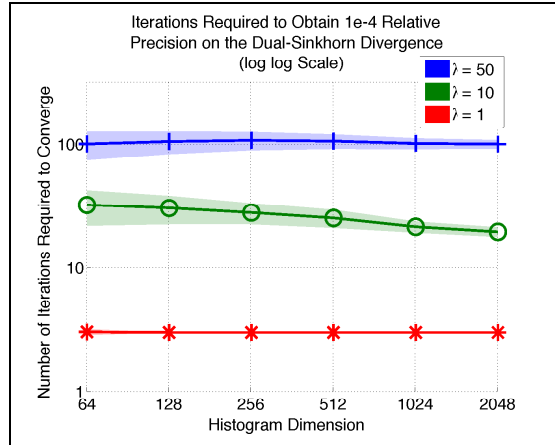

Figure 5: The influence of $\lambda$ on the number of iterations required to converge on histograms uniformly sampled from the simplex.

## 6 Conclusion

We have shown that regularizing the optimal transport problem with an entropic penalty opens the door for new numerical approaches to compute OT. This regularization yields speed-ups that are effective *regardless of any assumptions on the ground metric $M$*. Based on preliminary evidence, it

seems that dual-Sinkhorn divergences do not perform worse than the EMD, and may in fact perform better in applications. Dual-Sinkhorn divergences are parameterized by a regularization weight $\lambda$ which should be tuned having both computational and performance objectives in mind, but we have not observed a need to establish a trade-off between both. Indeed, reasonably small values of $\lambda$ seem to perform better than large ones.

**Acknowledgements**    The author would like to thank: Zaid Harchaoui for suggesting the title of this paper and highlighting the connection between the mutual information of $P$ and its Kullback-Leibler divergence to $rc^T$; Lieven Vandenberghe, Philip Knight, Sanjeev Arora, Alexandre d'Aspremont and Shun-Ichi Amari for fruitful discussions; reviewers for anonymous comments.

# 7    Appendix: Proofs

*Proof of Property 1.* The set $U_1(r,c)$ contains all joint probabilities $P$ for which $h(P) = h(r) + h(c)$. In that case (Cover and Thomas, 1991, Theorem 2.6.6) applies and $U_1(r,c)$ can only be equal to the singleton $\{rc^T\}$. If $M$ is negative definite, there exists vectors $(\varphi_1, \cdots, \varphi_d)$ in some Euclidean space $\mathbb{R}^n$ such that $m_{ij} = \|\varphi_i - \varphi_j\|_2^2$ through (Berg et al., 1984, §3.3.2). We thus have that

$$r^T M c = \sum_{ij} r_i c_j \|\varphi_i - \varphi_j\|^2 = (\sum_i r_i \|\varphi_i\|^2 + \sum_i c_i \|\varphi_i\|^2) - 2\sum_{ij} \langle r_i \varphi_i, c_j \varphi_j \rangle$$

$$= r^T u + c^T u - 2r^T K c$$

where $u_i = \|\phi_i\|^2$ and $K_{ij} = \langle \varphi_i, \varphi_j \rangle$. We used the fact that $\sum r_i = \sum c_i = 1$ to go from the first to the second equality. $r^T M c$ is thus a n.d. kernel because it is the sum of two n.d. kernels: the first term $(r^T u + c^T u)$ is the sum of the same function evaluated separately on $r$ and $c$, and thus a negative definite kernel (Berg et al., 1984, §3.2.10); the latter term $-2r^T K u$ is negative definite as minus a positive definite kernel (Berg et al., 1984, Definition §3.1.1). ∎

*Remark.* The proof above suggests a faster way to compute the Independence kernel. Given a matrix $M$, one can indeed pre-compute the vector of norms $u$ as well as a Cholesky factor $L$ of $K$ above to preprocess a dataset of histograms by premultiplying each observations $r_i$ by $L$ and only store $Lr_i$ as well as precomputing its diagonal term $r_i^T u$. Note that the independence kernel is positive definite on histograms with the same 1-norm, but is no longer positive definite for arbitrary vectors.

*Proof of Lemma 1.* Let $T$ be the a probability distribution on $\{1, \cdots, d\}^3$ whose coefficients are defined as

$$t_{ijk} := \frac{p_{ij} q_{jk}}{y_j}, \tag{3}$$

for all indices $j$ such that $y_j > 0$. For indices $j$ such that $y_j = 0$, all values $t_{ijk}$ are set to 0.

Let $S := [\sum_j t_{ijk}]_{ik}$. $S$ is a transport matrix between $x$ and $z$. Indeed,

$$\sum_i \sum_j s_{ijk} = \sum_j \sum_i \frac{p_{ij} q_{jk}}{y_j} = \sum_j \frac{q_{jk}}{y_j} \sum_i p_{ij} = \sum_j \frac{q_{jk}}{y_j} y_j = \sum_j q_{jk} = z_k \text{ (column sums)}$$

$$\sum_k \sum_j s_{ijk} = \sum_j \sum_k \frac{p_{ij} q_{jk}}{y_j} = \sum_j \frac{p_{ij}}{y_j} \sum_k q_{jk} = \sum_j \frac{p_{ij}}{y_j} y_j = \sum_j p_{ij} = x_i \text{ (row sums)}$$

We now prove that $h(S) \geq h(x) + h(z) - \alpha$. Let $(X, Y, Z)$ be three random variables jointly distributed as $T$. Since by definition of $T$ in Equation (3)

$$p(X, Y, Z) = p(X, Y)p(Y, Z)/p(Y) = p(X)p(Y|X)p(Z|Y),$$

the triplet $(X, Y, Z)$ is a Markov chain $X \to Y \to Z$ (Cover and Thomas, 1991, Equation 2.118) and thus, by virtue of the data processing inequality (Cover and Thomas, 1991, Theorem 2.8.1), the following inequality between mutual informations applies:

$$I(X;Y) \geq I(X;Z), \text{ namely } \quad h(X,Z) - h(X) - h(Z) \geq h(X,Y) - h(X) - h(Y) \geq -\alpha.$$

∎

## Footnotes

[1] $\exists n, \exists \varphi_1, \cdots, \varphi_d \in \mathbb{R}^n$ such that $m_{ij} = \|\varphi_i - \varphi_j\|_2^2$. Recall that, in that case, $M$ raised to power $t$ element-wise, $[m_{ij}^t], 0 < t < 1$ is also a Euclidean distance matrix (Berg et al., 1984, p.78,§3.2.10).

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
