[Reviews · NeurIPS 2013]

Submitted by Assigned_Reviewer_5

This paper proposes a new family of "transportation distances" for discrete domains, such as histograms, for which the earth movers' distance (EMD) has been widely used. The problem is that EMD is very expensive to compute as it is cubic (actually d^3 log d) in the dimension of the observation space (the number of histogram dimensions, d). In this paper it is argued that with the inclusion of an entropic regularizer finding the optimal transportation distance becomes a much simpler computational problem, one that can be solved in linear time and is easily parallelized.


I do not work with EMD or such measures extensively, but I very much enjoyed reading this paper. The formulation seems very clean and the result appear impressive. The problem is well motivate, the reader is guided through the technical development quietly nicely, and the exposition is generally very good. A nice paper to read, with novel results (to me at least), and the potential for significant impact in several areas of vision and learning.


The weakness, as such, is in the experiments. I would;t have thought that MNIST was a natural dataset for this technique. First the inputs are images which can be considered discrete inputs for which one wants to use EMD, but most people would place them in a continuous vector space. Also, while the ordering of bins in a histogram may have no intrinsic ordering per se, this is not the case with an image. And finally, since MNIST has been beaten mercilessly in the last couple of decades it is not surprising that the error rates on MNIST are not particularly good by most standards. That said, the results do strongly support the claims of the authors that the new optimal transportation distance is both effective and extremely efficient to compute with the what they refer to as the Sinkhorn-Knopp algorithm.
Summary: This paper proposes a new family of optimal transportation distances for histogram domains. By including an entropic regularizer to a conventional distance measure they show that one can compute the optimal transportation distance in linear rather than cubic time. This should make use of the EMD much more practical for a wide spectrum of tasks.


Submitted by Assigned_Reviewer_9

This manuscript proposes to regularize the computation of the earth
mover's distance (EMD) with an entropic term, with the dual aim of
improving the empirical performance of the distance and in
significantly speeding up its computation. The manuscript presents
theoretical analysis in support of this regularization approach, and
empirical results demonstrated improved classification accuracy for
regularized EMD relative to a variety of distances (including
classical EMD), as well as faster computation time.

The theoretical analysis, explaining how the entropic regularization
works, proving that it is metric, and describing how to compute it
efficiently, are interesting, though it is not clear how much of this
is truly novel, as opposed to being novel within the NIPS community.
The authors mention that this type of regularization has long been
known in the transportation theory literature.

The MNIST experiments are well done, and the results are compelling.
Similarly, the timing experiment shows convincingly that the speedup
achieved here is significant. What is missing is timing information
for a real classification problem. The manuscript should have at
least mentioned the relative timing of the computations for the MNIST
data sets.

Smaller comments:

"EMD" (p. 2) needs to be written out when it is first used.

The abbreviations for the key in Figure 2 need to be spelled out.

The experiment in Section 5.1 apparently uses a variety of kernels
within an SVM, but the SVM is only mentioned obliquely. The first
sentence in the section makes it sound like the distances are being
used directly to solve the MNIST problem.

What exactly is the confidence interval shown in Figure 2?
Confidences should be computed w.r.t. 6 complete labelings of the data
set, not 24 (i.e., the CV folds should not contribute to the
variance).

I don't understand why the manuscript says "Sinkhorn distances are
10.000 times faster than EMD solvers." Is the "." a decimal point?
Or is this supposed to be "10000 times faster"? Neither conclusion
seems to be justified by the results in Figure 4.

Summary: Introduces to NIPS readers a regularized earth
mover's distance. The concept is compelling, and the empirical
results (both in terms of classification performance and running
time) are convincing.

Submitted by Assigned_Reviewer_11

The paper studies variants of the Earth Mover's distance (EMD) with either a constraint on the entropy or a penalty term in the objective linear in the entropy. When there is a bound on the entropy, the objective turns out to be a distance metric but the paper does not show how to compute it efficiently. The other variant is not a distance metric but the paper shows that the optimal transportation is the unique stochastic matrix obtained by scaling rows and columns of the matrix with entries exp(-lambda*m_{i,j}) where m_{i,j} is the distance/transportation cost between i and j and lambda is the coefficient of the entropy term in the objective. This stochastic matrix can be approximated by Sinkhorn-Knopp's algorithm. The paper makes an interesting observation about a variant of EMD that might be efficiently approximated. However, the comparison with EMD in the paper is not fair, since the authors insist on EMD being computed exactly while the variant being studied is only approximated without any statement on the convergence rate of the algorithm. It is conceivable that for the random vectors tested in the paper, the algorithm converges quickly but it might be slow for pathological vectors. It is also not clear if comparing with EMD on just the original pixel space is the right comparison when the previous work cited by the paper computed EMD in the gradient orientation histograms (SIFT features) as well as the pixel space. The authors might also want to compare the results with previous work using variants/approximations of EMD such as
Kristen Grauman, Trevor Darrell. Approximate Correspondences in High Dimensions. NIPS 2006.
The pyramid match approach above is inspired by ideas from work on approximating EMD e.g.
Indyk, P., and Thaper, N. 2003. Fast Image Retrieval via Embeddings.

The paper is generally well written but the authors might want to include a statement of the Sinkhorn-Knopp's theorem, a description of the algorithm using formula rather than Matlab code, and some statement on the convergence rate of the algorithm.

============================

Rebuttal comments: there are many ways to speed up EMD computation, one such approach is to use coarser distances as taken by Pele and Werman already cited in the paper (round so that distances can only be 1,2,3). I am not sure if there are publicly available implementations taking advantage of the small weight range even though this is an area with a lot of theoretical results such as

Gabow, Harold N., and Robert E. Tarjan. "Faster scaling algorithms for network problems." SIAM Journal on Computing 18.5 (1989): 1013-1036.

It would be interesting to compare to that approach in terms of speed and retrieval accuracy.
Summary: The paper makes an interesting observation that after adding a penalty term based on the entropy of the transportation to the objective of EMD, the solution can be found via Sinkhorn-Knopp's algorithm. The variant seems interesting and might merit further investigation but the comparison made in the paper with EMD is inadequate.
Author Feedback

Author rebuttal: We thank the reviewers for their reviews. Some of the issues they raise are crucial, and we will incorporate this discussion in our draft. We also thank them for reading this rebuttal.

****** Reviewer 11:
- However, the comparison with EMD in the paper is not fair, since the author insists on EMD being computed exactly

Many thanks for this comment which underlines, in fact, a strength of our approach. We need to clarify this.

We do no insist on getting exact solutions for EMD: this is in fact the only practical choice we have. All EMD solvers that we know of build upon the network simplex. None offers an early stopping option.

Because of its combinatorial nature, the simplex does not have obvious stopping rules. Indeed, the change in the objective between two successive iterations is not necessarily decreasing, and one can build examples where the simplex achieves its largest improvement at the last iteration. For instance, in 3D, think of a feasible set shaped like an ice cream cone: a polyhedral ball on top of a cone. Assume the simplex starts from the top of the ice-cream ball and tracks back to the optimum, the tip of the cone, moving from vertex to vertex. The very last iteration gives the largest decrease in the objective.

We could imagine computing in parallel a dual simplex (this would basically double the execution cost) and use the dual gap as a stopping rule, or stop arbitrarily the execution. To our knowledge, neither has been studied nor advocated. This stands in stark contrast to the ability we have to stop early with the Sinkhorn method, and easily control this approximation by checking the convergence of x or that of the marginals of the current iterate.

- [...] without any statement on the convergence rate of the algorithm

Sinkhorn-Knopp has a linear convergence rate (l.57,l.367): an upper bound on |x_t-x^*| decreases geometrically with t. The constant is reported by Knight (2008) for the bistochastic case. Adapting these results for non-uniform marginals is straightforward (private communication with Knight) and we will add them. The constant depends on the smallest values of r_i, c_i and exp(-lambda m_ij) and converges to 1 as these numbers go to 0.

- It is also not clear if [...].

We chose the MNIST dataset because there is no ambiguity on what the ground metric should be (pixel distances) and because it is a widely recognized benchmark. We have started looking into more advanced datasets in computer vision.

- The authors might also want to compare [...]

We will. Note however that these approximations of EMD are known to perform worse than the exact EMD and can only compare clouds of points in low dimensions (e.g. planar case), where the ground metric M can only be the Euclidean distance between these points. Our approach works for any ground metric M and any Wasserstein exponent.

- might want to include [...]
We will.

****** Reviewer 5
- while the ordering of bins in a histogram may have no intrinsic ordering per se, this is not the case with an image.

Our histograms are not permutation invariant: for all 20x20 pixel intensities, namely d=400 bins in the grid, we use for m_ij the Euclidean distance between these two bins i and j (l.293). Please correct us if we have misunderstood your comment.

****** Reviewer 9:
- it is not clear how much of this is truly novel, as opposed to being novel within the NIPS community.

Thank you for pointing this out. Most of the paper is truly novel, let us draw the line:

What was known in the transportation literature & introduced here to the NIPS community:
= The regularized transportation problem can be computed using Sinkhorn-Knopp (a.k.a Bregman balancing) to obtain a "smoother" optimal transportation. Such regularized optima are used to predict the actual movements of people in transportation networks.

Our contributions:
= Using the solution of the regularized problem to define a transportation metric
= Proving that a hard threshold on the entropy of P results in a true metric (with a new version of the gluing lemma, lemma 1), and that alpha=0 yields a psd kernel.
= Vectorizing Sinkhorn-knopp: optimal transportations from one r to many c's can be computed altogether, with large speed-ups over separate executions. Specialists of the algorithm we have approached (Knight) were unaware of this fact. This remark is trivial in hindsight, but crucial to improve performance, use GPUs, and beat the simplex by a large factor.

- [...] relative timing of the computations for the MNIST data sets.

The MNIST dataset is such that d=400 but most vectors are sparse, typically of support 120-180. Numbers below are for one average execution:
emd_fast: 200 ms
emd_mex: same if naively used, but can be tweaked to leverage sparsity: 43 ms
Sinkhorn (lambda=9) 0.82 ms on a single core (50x faster than emd_mex).
SinkhornGPU (lambda=9) 0.30 ms (150x faster).

The speed up using GPUs here is less impressive (extra x3 instead of extra x10) because GPUs are not fast at indexing (extracting non-zero components). These numbers agree with figure 4 (for d between 128 & 256)

- the SVM is only mentioned obliquely [...]
Your question is answered in l.279. We will clarify this.

- the CV folds should not contribute to the variance
We were not aware of this issue, many thanks for pointing this out. Can you give us a reference please? Does your remark still apply in our setting where training sets do not overlap? (we only train on 1 fold an test on the 3 remaining folds)

- up to 10000 times faster
Indeed, we meant 10,000 and not 10.000. These speedups refer to the area for large d of figure 4. One unit of the log-scale grid is equal to a 10^2 speed up, so we do indeed observe that order of magnitude on the right side.

PS: there are two small typos in Algorithm 1. We apologize for the confusion:
l.235: Set x=ones(length(r),size(c,2))/length(r);
l. 238: v=c.*(1./(K'*u))